# Effects of Probiotics Administration on Human Metabolic Phenotype

**DOI:** 10.3390/metabo10100396

**Published:** 2020-10-07

**Authors:** Veronica Ghini, Leonardo Tenori, Marco Pane, Angela Amoruso, Giada Marroncini, Diletta Francesca Squarzanti, Barbara Azzimonti, Roberta Rolla, Paola Savoia, Mirko Tarocchi, Andrea Galli, Claudio Luchinat

**Affiliations:** 1Consorzio Interuniversitario Risonanze Magnetiche di Metallo Proteine (CIRMMP), 50019 Sesto Fiorentino, Italy; ghini@cerm.unifi.it; 2Magnetic Resonance Center (CERM), University of Florence, 50019 Sesto Fiorentino, Italy; tenori@cerm.unifi.it; 3Department of Chemistry, University of Florence, 50019 Sesto Fiorentino, Italy; 4Probiotical S.p.A., 28100 Novara, Italy; m.pane@probiotical.com (M.P.); a.amoruso@mofinalce.it (A.A.); 5Department of Experimental and Clinical Biochemical Sciences “Mario Serio”, University of Florence, 50100 Firenze, Italy; giada.marroncini@unifi.it (G.M.); mirko.tarocchi@unifi.it (M.T.); a.galli@dfc.unifi.it (A.G.); 6Department of Health Sciences (DiSS), University of Piemonte Orientale (UPO), Via Solaroli 17, 28100 Novara, Italy; diletta.squarzanti@med.uniupo.it (D.F.S.); barbara.azzimonti@med.uniupo.it (B.A.); roberta.rolla@med.uniupo.it (R.R.); paola.savoia@med.uniupo.it (P.S.); 7Center for Translational Research on Autoimmune and Allergic Diseases (CAAD), DiSS, UPO, Corso Trieste 15/A, 28100 Novara, Italy; 8Clinical Chemistry Unit, Azienda Ospedaliero Universitaria Maggiore della Carità, Corso Mazzini 18, 28100 Novara, Italy; 9SCDU Dermatology, AOU Maggiore della Carità, 28100 Novara, Italy

**Keywords:** probiotics, metabolomics, nuclear magnetic resonance spectroscopy, gut microflora

## Abstract

The establishment of the beneficial interactions between the host and its microbiota is essential for the correct functioning of the organism, since microflora alterations can lead to many diseases. Probiotics improve balanced microbial communities, exerting substantial health-promoting effects. Here we monitored the molecular outcomes, obtained by gut microflora modulation through probiotic treatment, on human urine and serum metabolic profiles, with a metabolomic approach. Twenty-two subjects were enrolled in the study and administered with two different probiotic types, both singularly and in combination, for 8 weeks. Urine and serum samples were collected before and during the supplementation and were analyzed by nuclear magnetic resonance (NMR) spectroscopy and statistical analyses. After eight weeks of treatment, probiotics deeply influence the urinary metabolic profiles of the volunteers, without significantly altering their single phenotypes. Anyway, bacteria supplementation tends to reduce the differences in metabolic phenotypes among individuals. Overall, the effects are recipient-dependent, and in some individuals, robust effects are already well visible after four weeks. Modifications in metabolite levels, attributable to each type of probiotic administration, were also monitored. Metabolomic analysis of biofluids turns out to be a powerful technique to monitor the dynamic interactions between the microflora and the host, and the individual response to probiotic assumption.

## 1. Introduction

The human gut normally hosts around 1000 different species of bacteria for a total of 1014 commensal microorganisms; approximately 1.3 times more than the number of eukaryotic cells constituting the human body [1,2,3]. The majority of these are strict anaerobes which outnumbers those facultative and aerobic by approximately 2–3 orders of magnitude [4].

The whole bacterial genome, the so-called intestinal microbiome (or “microbe” organ: 200 g in weight [5]), contains more than 5 million genes, a number that is 100-fold higher than the number of human genes [6].

The large arsenal of microbial genes provides a diverse range of biochemical and metabolic activities to complement host physiology and affect the hosts through huge benefits. The microbiota is essential for the development and correct functioning of the gut, helping the development of intestinal microvilli, and it is necessary for the development of the immune system. They are also involved in the digestion of polysaccharides and in the production of essential vitamins [7]. Moreover, they promote angiogenesis, regulate fat storage and protect against potential opportunistic pathogens [8]. So, the human body can be considered as a “superorganism”, with the gut microbiota collectively acting as a major virtual organ that contributes to host’s metabolism and physiology [9].

To ensure its crucial role in health promotion, the gut microflora composition normally undergoes constant and rapid changes, both in the structure and in the function, according to age, general conditions, lifestyle, diet habits, hygiene preferences or antibiotics intake [10]. Conversely, when the microflora is drastically altered, the well-being of many human niches can be compromised, leading to many chronic and degenerative diseases [11,12,13]. Accordingly, the establishment and preservation of beneficial interactions between the host and its associated intestinal microbiota are key requirements for an optimal psycho-physical state promotion. In this context, probiotics are increasingly used, with the final aim of manipulating the composition of the gut microbiota, improving balanced microbial communities, and thus exerting substantial health-promoting effects to the host [14,15,16,17]. Probiotics are “live microorganisms that, when administered in adequate amounts, confer a health benefit on the host” [17,18,19]. It must be underlined that, beyond composition, much of the impact of the bacterial communities on humans is carried out through the production and/or transformation of several metabolites, including amino acids, fatty acids, bile acids, hormones and vitamins [7]. Thus, in order to better understand how probiotic interventions influence the host, the analysis of the microbial communities should always be complemented by metabolomic analysis [20,21].

One of the main findings of metabolomics is that systemic biofluids like blood, urine and saliva, contain a strong individual metabolic signature, called metabolic phenotype or metabotype, which is stable over time, allowing us to monitor individual status and response to different stimuli [22,23,24,25,26,27], as well as to fingerprint several different diseases [28,29,30,31,32,33]. The metabotype is a multifactorial entity; in fact, besides being a snapshot of the subject’s genotype and metabolism, it is also influenced by many other factors, like gut microbiota, diet, lifestyle, drugs, etc. [34]. So-called bioactive food also impacts the human metabolome [35]. The influence of gut microbiota on the metabotype is particularly strong in the case of urine [36,37]. Under this light, the human metabolic phenotype could prove to be a very accurate and dynamic mirror, not only of the human organism, but, more correctly, of the “superorganism” and of its functioning and adapting processes [38,39,40].

In the present study, ^1^H nuclear magnetic resonance (NMR) spectroscopy has been used to explore and characterize the metabolic changes induced in the urine and serum metabotypes of 22 healthy adult volunteers by the regular administration of probiotics. 

^1^H-NMR-based metabolomics could be a powerful technique to gain a deep insight on the molecular mechanisms and effects of the dynamic interactions between the commensal microflora and the host, and of the response to probiotic assumption. 

The probiotics used in the present study were *Lactobacillus delbrueckii subs. delbrueckii* (LDD01 or L here after) and a mix of five biotypes of *Bifidobacterium longum* (DLBL07 to DLBL11, DLBL or B here after). The two different supplementations were singularly administered in the first 4–6 weeks of the experimental trial, and together in the subsequent 4–6 weeks. Univariate and multivariate statistical analyses have been used to evaluate the ^1^H-NMR urine and serum spectra acquired, and to characterize the individual effects of the treatment. 

## 2. Results

### 2.1. Study Design

The main features of the analyzed cohorts and collected samples are described in the Materials and Methods section and the Participation Flowchart is reported in Appendix A.

The study was based on three different time frames (Figure 1A):

Phase I (baseline, BL). A period of 4–6 weeks during which the volunteers did not take any probiotics. They proceeded with their usual diet and lifestyle without any standardization imposed. Twenty urine samples were collected from each volunteer (1 sample a day, excluding the weekend and the menstrual cycle days). At the end of phase I, a serum sample from each subject was also collected.

Phase II (single probiotic administration, SP). A period of 4-6 weeks during which the volunteers were randomly divided into two groups, named “L + LB” and “B + BL”. The “L + LB” group (10 subjects) added to their usual diet a daily dose of 1 billion of freeze-dried LDD01 probiotics, whereas the other group (“B + BL” group, 12 individuals) was supplemented with a daily dose of 1 billion of the DLBL mix. Again, 20 urine samples were collected as previously described. At the end of phase II, a serum sample from each subject was also collected.

Phase III (coupled probiotic administration (CP) or wash-out (WO)). In the last period of 4–6 weeks, 18 volunteers added both 1 billion of LDD01 and 1 billion of DLBL, together, to their usual diet. Thus, “L + LB” group (8 subjects) added 1 billion of DLBL to the daily dose of LDD01 and “B + BL” group (10 subjects) added 1 dose of LDD01 to the daily dose of DLBL. 

Three volunteers (two from the L + LB group and 1 from the B + BL group) performed this phase as a wash-out phase (WO) during which they did not assume any probiotics. Again, 20 urine samples were collected from each volunteer. At the end of phase III, a serum sample from each subject was also collected.

Twenty-two volunteers performed the first and the second phase of the study. Of these, 18 subjects completed the third phase as coupled probiotic administration, while three subjects as wash-out; one volunteer did not perform the third phase (Figure 1B).

### 2.2. Influence of Probiotic Treatment on Individual Urine Metabolic Phenotype

In 2015, our research group demonstrated that the metabolic phenotype, based on the NMR spectral profiles of urine samples, is specific to each individual and stable over a time frame of 10 years, in the absence of physio-pathological stressful conditions [24]. We also established that multivariate statistical analysis of multiple urine samples of individuals permits one to define their placement in the “metabolic space”, where each subject can be discriminated from the others with an accuracy close to 100% (97–98%) [22,23,24]. Here, we applied the same statistical analysis (principal component analysis (PCA)- canonical analysis (CA)- K-nearest neighbors (KNN)) used in our previous studies to characterize the urine individual metabolic phenotype of the volunteers, and to investigate if after probiotics assumption the individual phenotype significantly changes as a consequence of gut microflora modulation.

As expected [22,23,41], the daily intra-individual differences were by far smaller than the inter-individual separation, allowing individual discrimination: considering all samples collected in phase I, the individual discrimination was almost perfect, with an error of only 3% (Figure 2A). Phase I was designed to establish a baseline reference for each volunteer in comparison with phase II and III, where effects of probiotic administration could be visible.

The error in the individual discrimination increased slightly (6%) within samples collected during phase II (Figure 2B); however, the slight increase with respect to phase I is statistically significant. Interestingly, within samples collected in phase III, a further significant increase of the error in the individual discrimination up to 10% was observed (Figure 2C).

Thus, probiotics administration influences the individual metabotype: upon administration of the same probiotics, individuals moved closer to one another in the phenotypic space. Nevertheless, they remained well discriminated from each other, and the effects were not so strong as to hamper the individual recognition.

Considering “L + LB” group and “B + BL” group separately, the error in the individual discrimination using samples collected in phase I is always very low, around 3% for both (Figure 2D). For “L + LB” group, the individual discrimination error remained around 3% within phase II, whereas it increased considering samples of phase III (7%). On the contrary, in the case of “B + BL” group, the error in the individual discrimination increased from phase I (3%), to phase II (6%) and to phase III (11%) (Figure 2D). 

The increment of the discrimination error among volunteers seems to be related to the administration of DLBL more than LDD01. Subjects of “B + BL” group were administered with DLBL, both in phase II and in phase III, whereas in “L + LB” group the volunteers assumed DLBL only in phase III. According to this, in “B + BL” group, the error in the individual discrimination started to increase from phase II and the effect became higher in phase III, whereas, in “L + LB” group, the error in the discrimination increased only in phase III.

### 2.3. Individual Urine Effects of Probiotic Administration

To underline the overall outcomes of probiotics assumption, the urine samples, collected during the three phases of the study, were compared using multilevel partial least squares (M-PLS) analysis (Figure 3). A multilevel approach was chosen to suppress the noise due to intra-individual variability, i.e., each individual acts as control for him/herself. The resulting accuracy explains how much the groups (and so the individual urine profiles) changed after the treatments. The application of M-PLS analysis resulted in a modest discrimination between samples collected in phase I and II, with a discrimination accuracy value of 85%, considering all the individuals together (Figure 3A,C). The discrimination accuracy became high, 95%, considering the comparison between urine samples of phase I and phase III, and a better separation of the two groups was achieved (Figure 3B,C).

Furthermore, each subject has also been considered separately, and PCA-CA-KNN was used to identify individual behaviors because of the probiotic intake (Table 1). From the obtained results, it was possible to identify two different effects on metabolic phenotypes: (i) in some subjects (i.e., PRB09, PRB14, PRB13) the discrimination accuracies between phase I and phase III remained the same (or slightly increased) with respect to the comparisons between phase I and phase II; according to this, there is a weak or even no discrimination between the samples of phase II and phase III (accuracy around 55–70%). This could mean that, for these subjects, 4–6 weeks (time length of phase II) are enough to have marked effects of probiotics on the metabolic profiles; subsequent probiotics administration in phase III did not alter more the metabolic profile.

(ii) Contrarily, in other subjects (i.e., PRB21, PRB17, PRB10) the discrimination accuracies gradually increased from the comparison between phase I vs. phase II and phase I vs. phase III. For these subjects, more time seemed to be needed to stabilize the metabolic effects of probiotic assumption, and phase III is crucial to have marked effects. 

In both cases, the discrimination accuracies for the comparison of phase I vs. phase III were very high for most of the volunteers. These data underlined that the effects of the treatment were strongly individual-dependent.

PRB06, PRB18, PRB24 performed phase III as a wash-out. Looking at their values of discrimination accuracies (Table 1), it can be concluded that the spectra of the urines collected in phase III remained more or less similar to those collected during phase II. It is possible to speculate that 4–6 weeks (time length of wash-out) were not enough to revert the effects introduced from probiotics assumption during phase II, thus suggesting a possible long-lasting intestinal colonization by the probiotic microorganisms.

To better illustrate the individual effects on metabolic profiles introduced by the daily consumption of probiotics, for each individual, the 20 spectra of urine samples collected in phases I, II and III, were considered as originating from different individuals (pseudo-individuals) [23]; therefore, each subject gives rise to three different pseudo-individuals (Figure 4).

Forty-four pseudo-individuals were created for the comparison of phase I vs. phase II. PCA-CA-KNN was used to determine the individual discrimination accuracies and errors and the confusion matrix is reported in Figure 4A. Again, two different behaviors can be detected. Some pseudo-individuals (belonging to the same volunteer) are relatively well discriminated from one another, with a discrimination accuracy of up to 75% (and an error less than 25%); they were recognized as two distinct individuals, indicating that the probiotic treatment introduced effects on their metabolic profiles. On the contrary, other pairs of pseudo-individuals were not discriminated from one another (discrimination error around 40–50%); they were recognized as the same individual, indicating that a period of 4–6 weeks of probiotics assumption (phase II) was not sufficient to establish clear differences. It is to be noted that these values of discrimination accuracies agree with those of discrimination accuracies reported in Table 1.

For the comparison between phases I and III, 42 pseudo-individuals were created (42, and not 44, pseudo-individuals were created because subject “PRB 23” had been eliminated from the analysis because he/she did not undergo the phase III probiotic administration). In this comparison, most pseudo-individuals couples originating from the same individual were very well discriminated from one another (Figure 4B). It is possible to note that, in this last case, the pseudo-individuals that have the lowest values of discrimination errors (highest values of discrimination accuracy) are those created with samples collected in phase I. On the contrary, the corresponding pseudo-individuals created with samples collected during phase III have higher values of discrimination error between one another. These data agree with the higher error in the individual discrimination obtained using only urine spectra belonging to samples collected in phase III (Figure 2).

### 2.4. Urine Metabolite Analysis

A set of 40 urine metabolites, whose peaks in the spectra were well defined and isolated, were selected and their level analyzed.

Five metabolites resulted to be significantly different (*p* < 0.05), at least in one pairwise comparison among phase I, phase II and phase III, considering all the individuals together (Table 2). Citrate, creatinine and 1-methylnicotinamide (mNam) were among the metabolites that showed significant changes (*p* < 0.01). Specific effects attributable to the type of probiotic have been also detected analyzing the subjects of “L + LB” group and “B + BL” group separately (Table 2).

The citrate concentration, whose levels correlate with those of calcium-citrate complexes, metabolic acidosis and the reduction of calcium/magnesium ions, increased in phase II in “L + LB” group; these higher levels were maintained (or even increased) also in phase III. Creatinine concentration, the degradation product of creatine, significantly increased in phase II only in “L + LB” group where the individuals were supplemented with LDD01. In this group, in phase III, creatinine concentration remained stable at the highest levels reached in phase II. Instead, in “B + BL” group, the concentration of this metabolite increased in a significant way only in phase III, when the volunteers added a daily dose of LDD01 together with the assumption of DLBL. So, the higher levels of creatinine seem to be attributable more to the effects of LDD01 administration rather than of DLBL. Moreover, in both groups, the levels of the niacin microbial metabolite mNam increased in phase II, but they became higher, and statistically significant, only in phase III, where the volunteers were administered with both probiotics together. This effect was stronger in “L + LB” group rather than in “B + BL” group.

In “L + LB” group, leucine, an essential amino acid for human well-being, presented higher concentration values in phase II than in phase I but, when the volunteers were administered with both probiotics together (phase III), its level decreased toward the reference value of phase I. This metabolite had the same trend during the three phases in “B + BL” group, but not in a significant way. Finally, in the same group, formate, key metabolite at the interface between the host and the microbiome metabolism, decreased its concentration in phase III.

### 2.5. Serum Metabolite Analysis

^1^H-NMR NOESY (Nuclear Overhauser Effect Spectroscopy) spectra of serum samples, collected at the end of each phase, were compared using M-PLS to check overall effects of probiotics assumption in blood. As in the case of urine, a multilevel approach has been used to suppress the noise introduced by the intra-individual variability [42]. In the case of serum samples, no discrimination was obtained between phase I and II (55%) (Figure 5A). The discrimination slightly increased (accuracy of 62%) when comparing samples belonging to phase I and phase III (Figure 5B).

Twenty-four metabolites, that are normally present in serum, and whose peaks in the CPMG spectra were well defined and isolated, were assigned and analyzed. Five of them resulted with significantly different concentration levels (*p* < 0.05), in at least one pairwise comparison among phase I, phase II and phase III (Table 3). Among them, pyruvate, phenylalanine and proline were the metabolites that showed the lowest *p*-values (*p* < 0.02) in both groups.

Pyruvate was the only metabolite whose concentration tended to increase during both phase II and phase III; while the increment is not statistically significant between phase I and II, it became more evident and significant when comparing phase I and III. Looking at the two groups separately, in “L + LB” group, the increment of the pyruvate levels, even not statistically significant, is visible only in phase III where the volunteers were administered with both probiotics (data not shown), whereas in “B + BL” group, the increased levels of pyruvate started in phase II and increased significantly during phase III. From these data, it is possible to note that higher levels of pyruvate are more related to the daily administration of DLBL rather than of LDD01. In “B + BL” group, proline and phenylalanine concentration levels tend to increase from phase I, to phase II; in phase III, the two metabolites increased significantly with respect to phase I. Despite not being statistically significant, the trend to increase of these metabolites has also been detected in “L + LB” (data not shown).

## 3. Discussion

The aim of the present study was to monitor the molecular outcomes obtained by gut microflora modulation through probiotic treatment, on human urine and serum metabolic profiles with a ^1^H-NMR-based metabolomic approach. To achieve this purpose, the human metabolome of 22 healthy individuals was monitored during probiotics administration.

We showed that the probiotic supplementation could influence the urine metabolic profiles of the volunteers, without altering in a substantial way the metabotypes of each individual.

It was previously demonstrated that obtaining a strong predictive model for the individual metabolic phenotype relies on the analysis of a sufficient number of urine samples from the same individual collected on different days. Twenty urine samples are enough to minimize the “metabolic noise” of the day-to-day variability and achieve a very good individual discrimination, with an error around 2–3%. A lower error, about 1%, can be obtained using as many as 40 urine samples per person [22,23,24]. Similar results can be obtained using multiple urine samples collected in the same day (4–5 per day, for 10 days) [27]. Interestingly, using the spectra of samples collected in phases II and III, the individual discrimination errors (6% and 10%, respectively) were significantly higher than those obtained and expected using samples collected in phase I (3%), where no effects attributable to probiotics assumption were present (Figure 2). This effect seems to be stronger in “B + LB” group, where individuals were administered with DLBL from the beginning of phase II (Figure 1).

Our data could be interpreted as an indication that the urine metabotype building blocks arising from the activity of the microbiome contribute to 8–10% of the intra-individual metabotype variability; this component tends to become similar among all the volunteers after 8–12 weeks of daily consumption of probiotics, slightly decreasing intra-individual variations. The effect of probiotic assumption (in particular for DLBL) seems to affect the metabolic phenotypes, but not so strongly as to hamper the individual recognition. This hypothesis was also confirmed by the pseudo individuals’ analysis (Figure 4).

By suppressing the intra-individual variability using a multilevel approach, it was possible to detect general effects of probiotics assumption in the urine spectra: samples collected during phase I were discriminated from samples collected during phase II, with an accuracy of 85%. The accuracy value became 95% considering the comparison between phase I and III (Figure 3). This means that, after 8–12 weeks, the effects of probiotics assumption become stronger and more detectable. This can be due to three different and interacting reasons: (i) build-up of probiotic abundance in time during the supplementation (from 4–6 weeks in phase II to 8–12 weeks in phase III), (ii) higher daily dosage (from 1 billion of probiotics per day in phase II, to two billion per day in phase III), (iii) possible synergistic interactions between LDD01 and DLBL. Anyway, the effects are individual-dependent. In some individuals, robust effects are already well visible at the end of phase II, after the daily assumption of LDD01 or DLBL; for others, instead, strong effects are visible only in phase III. Even in the case where the effects were well visible at the end of phase II, phase III further contributed to increase the accuracy for the recognition between pre- and post-probiotic assumption.

Modifications in metabolite levels, attributable to the administration of probiotic, could be due to a different food degradation and digestion by the bacteria, to the increased availability of metabolites, or to a direct effect of bacteria on the gut. Some of the alterations can be foreseen as positive effects on general health; others could be considered as dangerous, others of uncertain effect. In urine spectra, citrate, creatinine and mNam were among the metabolites that showed significant changes (*p* < 0.01) after the probiotic assumption (Table 2). 

Citrate in both groups increased in phase II and remained at a higher level in phase III. This metabolite in the urine has a very important role due to its inhibitory action in calcium stone formation [43]. Patients with calcium stones are characterized by low urinary citrate excretion; the normalization of citrate urinary concentration in this case resulted in a significant inhibition of crystal growth [44,45].

mNam has been found to increase especially in phase III; the assumption of LDD01 could be considered the main factor responsible of this phenomenon, because mNam was higher in volunteers who had been administered with LDD01 for a longer time (“L + LB” group). Until recently, mNam has been thought to be a biologically inactive product of nicotinamide metabolism (vitamin B3) in the pyridine nucleotides pathway [46]. Increased levels of this metabolite have also been found in the urine of adult mice treated with *Lactobacillus acidophilus* La5 and *Bifidobacterium lactis* Bb12 [47]. mNam is involved in the pathway of NAD synthesis, and an increase of NAD production can prevent age-related diseases [47].

Our data show that LDD01 supplementation increased the excretion rate of creatinine, a urinary metabolite derived from creatine. In fact, in “L + LB” group, higher levels of creatinine were present in samples collected during phases II and III; in “B + BL” group, an increase of this metabolite was detected only in phase III. Higher levels of creatinine and urea in blood can be considered as markers of renal failure. In fact, several in vivo and in vitro studies show that probiotics, especially mixtures of Lactobacilli, are able to decrease blood levels of urea, creatinine and other markers of kidney diseases [48].

Serum (or plasma), at variance with urine, is less affected by circadian variations. Due to this important systemic role, in the blood, the metabolite concentrations are strongly controlled through all cycles used for setting and maintaining equilibrium conditions. For these reasons, although some effect and interaction of the microbiome has also been characterized on blood metabolites and biochemistry [49], strong effects on the serum multivariate profile, attributable to probiotics administration, were not expected. Although no discrimination has been obtained in the multivariate analysis, that univariate revealed some metabolite changes in the serum of the volunteers during the study. In serum spectra pyruvate, phenylalanine and proline were the metabolites with significant changes (*p* < 0.02) in both groups (Table 3). Phenylalanine and proline showed the same trend along the three phases: in both groups, they increased from phase I to phase II, and the higher levels were maintained also during phase III. Phenylalanine is a potent cholecystokinin releaser, and it can slow down gastric emptying. High levels of this amino acid were also found in plasma mice after the administration of *Lactobacillus rhamnosus* [50]. In the same study, also proline, a non-essential amino acid in mammals with several key functions for human health [51], was increased due to the action of probiotics. Moreover, using in vivo approaches, after the assumption of *L. rhamnosus*, proline level became normal in alcoholic fatty liver disease mice [52].

The levels of pyruvate in sera increased after probiotic assumption in both groups. Pyruvate performs many functions in human metabolism, and it is an important intermediate in the production of ATP. Some pathologies, mostly neurodegenerative, involve an unbalanced pyruvate metabolism [53]. We can speculate that the use of these probiotic strains could be beneficial for the modulation of pyruvate excretion in these kinds of diseases.

The systematic assumption of probiotics can modulate some important functions on humans that have to be understood more deeply. Several in vivo studies have shown that probiotics can modulate the gut microflora in order to promote health and prevent, in particular, gastrointestinal and renal diseases. In humans, the situation is more complicated due to different genetic backgrounds and different metabolic variations. In fact, our results show that the response to probiotics is strongly individual-dependent and it could be related to genetic and environmental factors. Some metabolites that are mainly linked to the prevention and treatment of gastrointestinal and renal diseases and to keeping cells active by increasing biosynthesis of ATP and proteins [54] change in a significant way as a consequence of specific probiotics administration.

To our knowledge, this is one of the first studies combining the effects of probiotic administration on healthy human metabolism, with the possibility to characterize these effects by using ^1^H-NMR spectroscopy.

## 4. Materials and Methods

### 4.1. Recruitment of Volunteers and Exclusion Criteria

The trial was registered at clinical.trials.gov with the registration number NCT04506385. Twenty-two adult healthy individuals, 9 females and 13 males, between 24 and 62 years of age were recruited for this study. Exclusion criteria included previous surgery on the intestinal tract, probiotic/prebiotic and antibiotic treatments within 3 and 1 months, respectively, before the beginning of the study. In addition, subjects were excluded from the study if they were going to drastically change their diet or lifestyle during the study.

All subjects gave their informed consent for the inclusion in the study. The study was conducted in accordance with the Declaration of Helsinki, as revised in 2013. Ethical approval (protocol n° 294/CE, study n° CE 14/20, International Ethics Committee A.O.U. “Maggiore della carità”, Novara, Italy) or the study protocol was obtained.

### 4.2. Probiotic Formulation

The probiotic formulation named “LDD01” (batch R&S 0342-03, Probiotical S.p.A, Novara, Italy) contained a freeze-dried mixture > 1 × 10^9^ CFU/sachet *L. delbrueckii* subsp. *delbrueckii* MB386 (LDD01; DSM 22106) and 2.5 g of maltodextrin. The probiotic formulation named “DLBL” (batch R&S 0342-01, Probiotical S.p.A., Novara, Italy) contained a freeze-dried bacterial mixture > 1 × 109 CFU/sachet of *B. longum* DLBL07 (DSM 25669; Leibniz Institute DSMZ- German Collection of Microorganisms and Cell Cultures, GmbH), *B. longum* DLBL08 (DSM 25670), *B. longum* DLBL09 (DSM 25671), *B. longum* DLBL10 (DSM 25672), *B. longum* DLBL11 (DSM 25673) and 2.5 g of maltodextrin. Probiotic formulations were stored at 4 °C until use.

### 4.3. Sample Collection

All the samples were collected and processed according to standard operating procedures (SOPs) to obtain high quality samples for metabolomics [55,56,57]. Both serum and urine were collected under fasting condition. For urine, the midstream of the first urine of the morning was collected. All the processing procedures are detailed in Takis at al. [20]. During the course of the study, the samples were stored at −80 °C in the repository of the Da Vinci European Biobank, which offered a conservation service (daVEB, DOI:10.5334/ojb.af, https://www.unifi.it/vp-11370-da-vinci-european-biobank.html, Italy) [58,59]. 

### 4.4. NMR Analysis and Spectral Processing

NMR samples were prepared according to SOPs for urine and serum [20,21,60]. All the procedures for NMR sample preparation and analysis are detailed in Vignoli et al. [21]. Briefly, samples were analyzed using 4.25 mm NMR tube (Bruker BioSpin Gmbh, Rheinstetten, Germany). One-dimensional (1D) ^1^H-NMR spectra were acquired using a spectrometer operating at 600 MHz (Bruker BioSpin) optimized for metabolomics, equipped with a 5 mm triple resonance inverse (TCI) cryoprobe (CP) ^1^H-^13^C-^31^P and ^2^H-decoupling cryoprobe, including a z axis gradient coil, an automatic tuning-matching and an automatic sample changer. Samples were kept for 3 min inside the NMR probe head for temperature equilibration (300 K for urine and 310 K for serum). ^1^H-NMR spectra were acquired with water peak suppression and (i) a standard NOESY pulse sequence, for both urine and serum samples; (ii) a standard Carr–Purcell–Meiboom-Gill (CPMG) pulse sequence, for serum samples. 

Free induction decays were multiplied by an exponential function equivalent to a 1.0 Hz line-broadening factor, before applying Fourier transform. Transformed spectra were automatically corrected for phase and baseline distortions and calibrated to the reference signal of TMSP at δ 0.00 ppm, and to the glucose doublet at δ 5.24 ppm, for urine and serum respectively, using TopSpin 3.5 (Bruker BioSpin Gmbh, Rheinstetten, Germany).

### 4.5. Statistical Analysis

Various kinds of multivariate statistical techniques were applied on the bucketed spectra. Bucketing was performed by segmenting each spectrum in the range between 0.2 and 10.00 ppm into 0.02-ppm chemical shift regions (region between 6.0 and 4.5 ppm containing residual water signal was excluded). The corresponding spectral areas were integrated using AMIX software (Bruker BioSpin Gmbh, Rheinstetten, Germany). Individual-dependent probabilistic quotient normalization [32] was carried out on the obtained buckets prior to statistical analyses.

Principal component analysis (PCA) was used to obtain a preliminary outlook of the data (visualization in a reduced space, clusters detection, screening for outliers). Partial least squares (PLS) and multilevel PLS (MPLS) [42] were employed to perform supervised data reduction and classification; canonical analysis (CA) was used in combination with PCA to increase the supervised separation of the analyzed groups. Accuracy, specificity and sensitivity were estimated according to standard definitions. The global accuracy for classification was assessed by means of a Monte Carlo cross-validation scheme. 

Univariate analyses of the NMR data were performed on Fourier transformed and calibrated spectra. Metabolites originating well defined and resolved peaks in the spectra were assigned and their levels were analyzed. The assignment was performed using an ^1^H-NMR spectra library of pure organic compounds, public databases, e.g., Human Metabolome Database, storing reference ^1^H-NMR spectra of metabolites, spiking ^1^H-NMR experiments and using literature data. The relative concentrations of each metabolite were calculated by integrating the signals in the spectra. The non-parametric Wilcoxon-Mann-Whitney test was used for the determination of the meaningful metabolites that changed in concentration during each of the three phases. False discovery rate correction was applied using the Benjamini and Hochberg method [61] (FDR): an adjusted *p*-value of 0.05 was deemed significant. 

## Figures and Tables

**Figure 1 metabolites-10-00396-f001:**
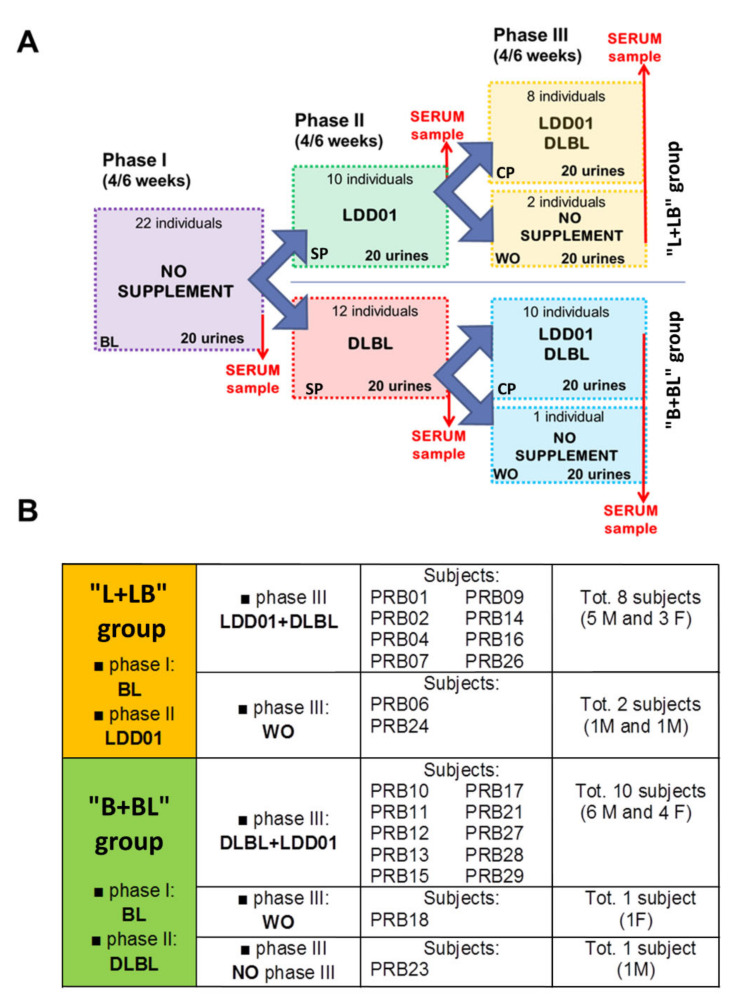
Experimental scheme. (**A**) Experimental scheme of the three phases of the project. (**B**) Table of the subjects that participated in the project in “L + LB” group and in “B + BL” group.

**Figure 2 metabolites-10-00396-f002:**
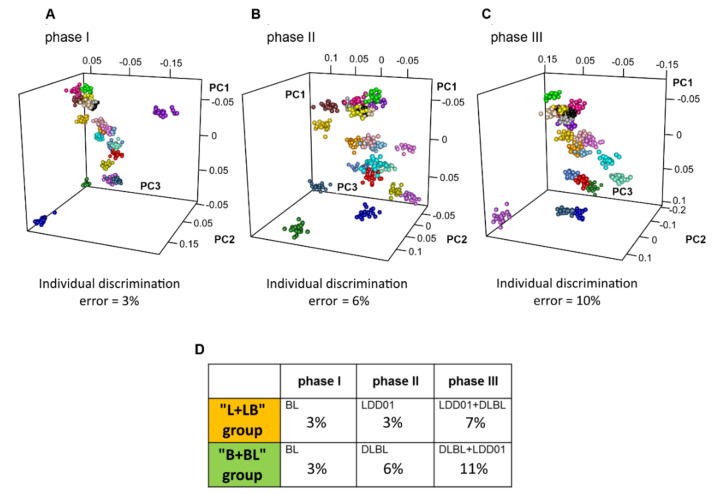
Urinary individual metabolic phenotype of the volunteers in (**A**) phase I (**B**) phase II (**C**) phase III. The score plots of PCA-CA-KNN individual discrimination are reported: each color represents a different volunteer, while each dot of the same color represents a different urine sample belonging to the same volunteer. PRB01: aquamarine; PRB02: red; PRB04: deep pink; PRB06: yellow; PRB07: light sky blue; PRB09: green; PRB10: purple; PRB11: gray; PRB12: orange; PRB 13: black; PRB14: gold; PRB15: cornflower blue; PRB16: brown; PRB17: light pink; PRB18: medium blue; PRB21: salmon; PRB23: dark yellow; PRB24: steel blue; PRB26: orchid; PRB27: dark orchid; PRB28: cyan; PRB 29: forest green. (**D**) Values of individual discrimination errors (PCA-CA-KNN individual discrimination), for “L + LB” group and “B + BL” group, separately, in phase I, in phase II and in phase III.

**Figure 3 metabolites-10-00396-f003:**
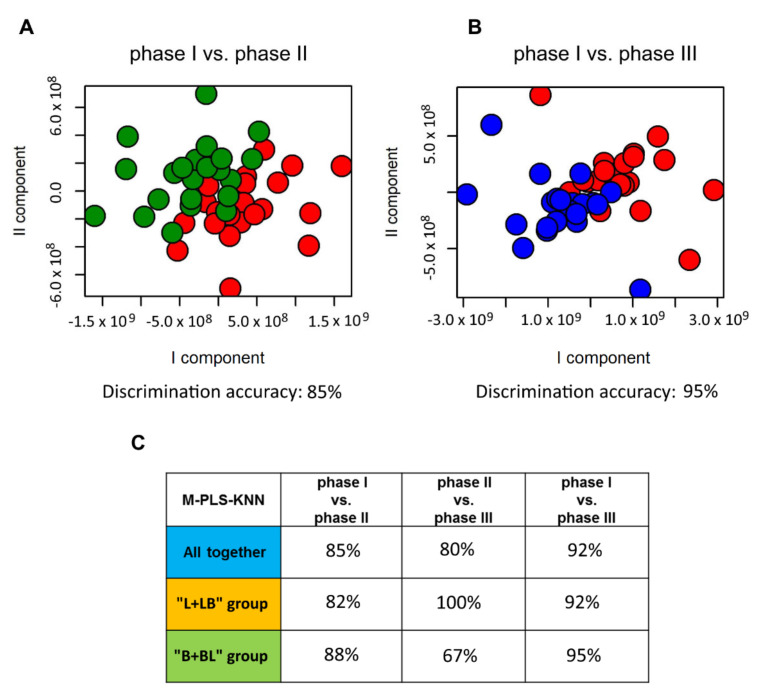
Score plots of M-PLS discrimination of the comparison between urine samples collected during (**A**) phase I (red spheres) and phase II (green spheres); (**B**) during phase I (red spheres) and phase III (blue spheres). (**C**) Discrimination accuracy values for the three pairwise comparisons among phase I, phase II and phase III, for all samples together, “L + LB” group and “B + BL” group. The median spectrum of each subject in each phase was calculated and used to build the MPLS models.

**Figure 4 metabolites-10-00396-f004:**
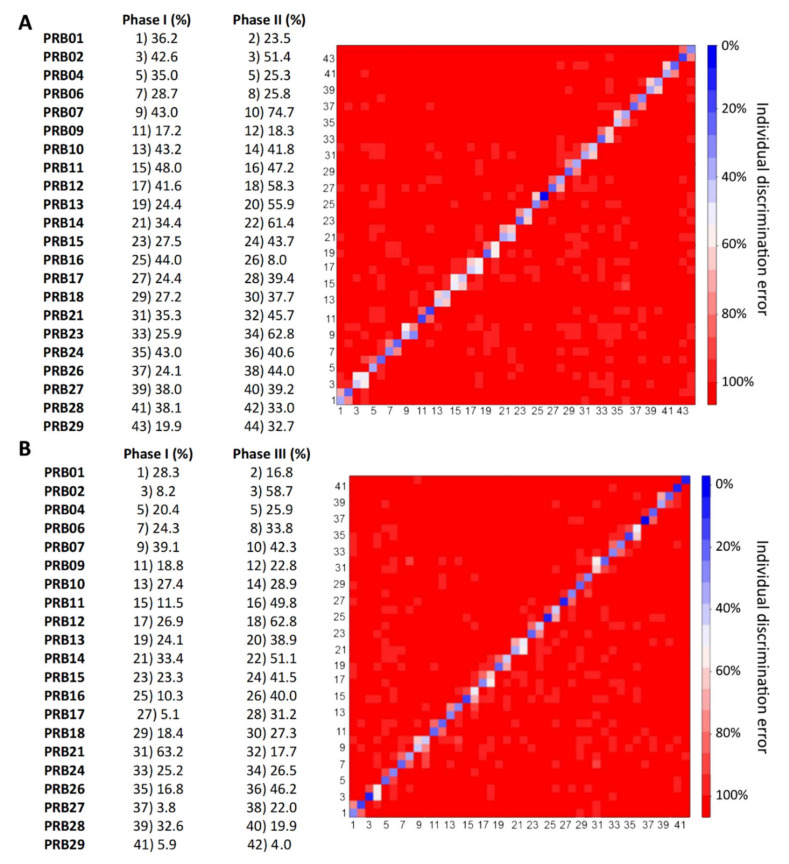
(**A**) Level plot of the confusion matrix of PCA-CA-KNN discrimination of 44 pseudo-individuals created for the comparison between phase I and II. (**B**) Level plot of the confusion matrix of PCA-CA-KNN discrimination of 42 pseudo-individuals created for the comparison between phase I and III.

**Figure 5 metabolites-10-00396-f005:**
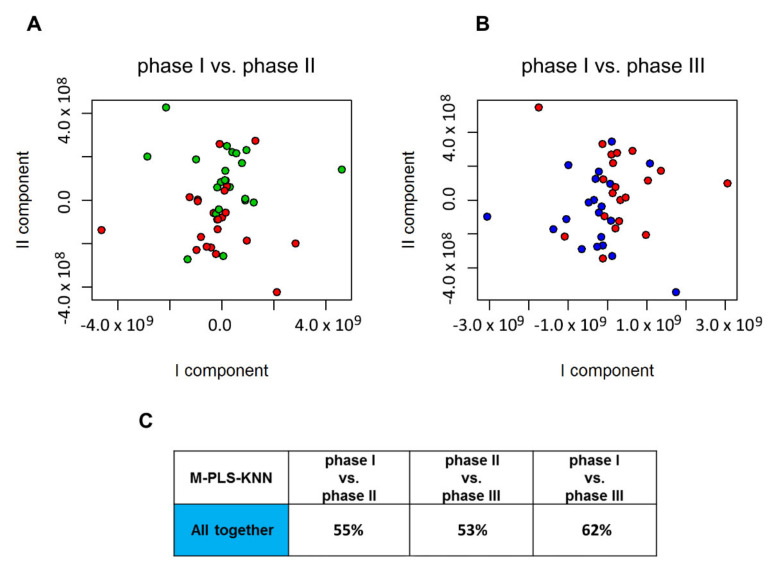
Score plot of M-PLS discrimination of serum samples collected during (**A**) phase I (red spheres) and phase II (green spheres); and (**B**) phase I (red spheres) and phase III (blue spheres). (**C**) Discrimination accuracy values for the three pairwise comparisons among phase I, phase II and phase III.

**Table 1 metabolites-10-00396-t001:** PCA-CA-KNN discrimination values for all the pairwise comparisons among phase I, phase II and phase III, for all the subjects individually.

“L + LB” Group				“B + BL” Group			
	Phase I vs. Phase II	Phase II vs. Phase III	Phase I vs. Phase III		Phase I vs. Phase II	Phase II vs. Phase III	Phase I vs. Phase III
**Subject:**	%	%	%	Subject:	%	%	%
**PRB01**	78	72	81	PRB10	65	68	85
**PRB02**	68	60	83	PRB11	73	83	85
**PRB04**	80	84	95	PRB12	58	50	68
**PRB06**	88	65 *	91 *	PRB13	83	61	92
**PRB07**	63	57	82	PRB15	75	67	75
**PRB09**	83	59	85	PRB17	84	72	99
**PRB14**	80	62	89	PRB18	78	73 *	86 *
**PRB16**	85	67	85	PRB21	72	74	89
**PRB24**	78	57 *	83 *	PRB23	70	--	--
**PRB26**	65	62	82	PRB27	56	73	80
				PRB28	68	74	73
				PRB29	74	98	97

Individuals that performed phase III as a wash-out are marked with *.

**Table 2 metabolites-10-00396-t002:** Urine metabolites that showed significant concentration changes at least during one phase of the project.

All Together	Phase I vs. Phase II	FC	Phase II vs. Phase III	FC	Phase I vs. Phase III	FC
Leucine	0.020	0.10	0.656	−0.03	0.222	0.07
Isoleucine	0.057	0.11	0.351	0.03	0.004	0.13
Valine	0.041	0.11	0.945	−0.01	0.079	0.10
Citrate	0.040	0.10	0.731	0.04	0.013	0.14
Creatinine	0.041	0.07	0.374	0.01	0.003	0.08
Choline	0.228	0.01	0.521	−0.06	0.013	−0.05
1-methylnicotinamide	0.185	0.03	0.083	0.01	6.32 * 10^−5^	0.04
Malonate	0.001	0.05	0.351	0.01	1.27 * 10^−5^	0.06
Valerate	0.040	0.03	0.858	−0.02	0.079	0.02
2-hydroxyisobutyric	0.358	0.14	0.351	−0.01	0.019	0.13
Acetoacetate	0.042	0.06	0.771	0.04	0.032	0.1
Fucose	0.178	0.05	0.374	0.02	0.021	0.07
Induxylsulfate	0.180	0.20	0.374	−0.12	0.004	0.08
Methanol	0.239	-0.1	0.008	−0.21	6.42 * 10^−5^	−0.31
Dimethylglycine	0.013	0.25	0.848	−0.04	0.008	0.21
Acetone	2.71 * 10^−5^	0.25	0.351	−0.09	0.013	0.16
Glycolate	0.042	0.46	0.752	−0.15	0.309	0.31
Ornithine+arginine-acetate	0.021	0.10	0.666	−0.03	0.001	0.07
4-hydroxyhippuricacid	0.586	0.17	0.376	0.14	0.041	0.31
**L + LB Group**	**Phase I** **vs.** **Phase II**	**FC**	**Phase II** **vs.** **Phase III**	**FC**	**Phase I** **vs.** **Phase III**	**FC**
Leucine	0.026	0.12	0.622	−0.05	0.359	0.07
Valine	0.025	0.16	0.622	−0.05	0.152	0.1
Citrate	0.071	0.09	0.466	0.07	0.012	0.16
Creatinine	0.071	0.07	0.622	0.03	0.042	0.010
Choline	0.135	0.03	0.466	0.03	0.001	0.06
1-methylnicotinamide	0.386	0.004	0.466	0.04	0.012	0.045
Malonate	0.059	0.06	0.925	−0.001	0.043	0.05
Induxylsulfate	0.984	0.036	0.466	0.18	0.043	0.22
Dimethylglycine	0.105	0.15	0.798	0.07	0.044	0.22
Acetone	0.021	↑0.22	0.622	−0.02	0.127	0.19
Glycolate	0.026	↑0.74	0.466	−0.5	0.273	0.24
Ornithine + arginine + acetate	0.091	0.09	0.834	−0.004	0.043	0.1
**B + BL Group**	**Phase I** **vs.** **Phase II**	**FC**	**Phase II** **vs.** **Phase III**	**FC**	**Phase I** **vs.** **Phase III**	**FC**
Isoleucine	0.282	0.12	0.282	0.02	0.033	0.14
Creatinine	0.281	0.07	0.282	0.003	0.036	0.07
Formate	0.860	0.08	0.860	−0.01	0.033	0.07
1-methylnicotinamide	0.517	0.05	0.517	−0.01	0.009	0.04
Malonate	0.038	0.04	0.038	−0.01	0.0001	0.03
Methanol	0.281	−0.1	0.282	−0.25	0.0001	−0.36
Acetone	0.005	0.28	0.005	−0.16	0.086	0.11
Erythritol	0.158	0.27	0.158	0.50	0.009	0.77
Ornith-arginine-acetate	0.236	0.11	0.236	−0.05	0.032	0.06
4-hydroxyhippuricacid	0.302	0.24	0.302	0.13	0.033	0.37

In the table the *p*-values obtained using false discovery rate (FDR) correction method and log_2_ (fold change) (FC) values are reported. Positive FC: increment; negative FC: decrement (in the second group of each comparison with respect to the first).

**Table 3 metabolites-10-00396-t003:** Serum metabolites with significant concentration changes at least during one phase of the project.

All Together	Phase I vs. Phase II	FC	Phase II vs. Phase III	FC	Phase I vs. Phase III	FC
proline	0.018	0.32	0.886	0.02	0.014	0.34
pyruvate	0.078	0.41	0.292	0.25	0.0006	0.65
dimethylglycine	0.046	−0.33	0.755	0.20	0.656	−0.13
phenylalanine	0.018	0.23	0.806	0.001	0.0006	0.23
**B + BL Group**	**Phase I** **vs.** **Phase II**	**FC**	**Phase II** **vs.** **Phase III**	**FC**	**Phase I** **vs.** **Phase III**	**FC**
pyruvate	0.209	0.46	0.762	0.31	0.033	0.77
phenylalanine	0.209	0.28	0.711	−0.003	0.038	0.28

*p*-values, obtained using FDR correction method and log_2_ (fold change) (FC) values are reported. Positive FC: increment; negative FC: decrement (in the second group of each comparison with respect to the first).

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
