# Peer review of "Effects of Probiotics Administration on Human Metabolic Phenotype"

_metabolites, 2020, doi:10.3390/metabo10100396_

Round 1
Reviewer 1 Report
Comments to the Authors
Ghini V. et al. reported on the “Effects of probiotics administration on human metabolic phenotype” (manuscript number: metabolites-941600).
The authors have shown that NMR-based biofluid metabolomic analysis is a powerful technique for monitoring dynamic interactions between the microbiota and the host, as well as individual responses to probiotic assumptions.
There is no fundamental criticism about the experimental approach. The manuscript is well and concisely written. Please see specific comments below.
Comment 1
It may be difficult due to the large number of subjects, but is it possible to show which subject's sample each plot is in Figures 1A-C?
Please show each color corresponding to the subject number in the Figure 1.
Comment 2
Tables 2 and 3 should show variable values (fold change) as well as p-values for each metabolite. It is important to show how the administration of probiotics altered urinary and serum metabolites.
Minor comments
Lines 123, 126, 129, 308, 343, 370, 372, 378, 414, and 416: check if the half-width is free (e,g., change “phenotypeSubsection” to “phenotype Subsection”).
Line 306: change “3-2%” to “2-3%”.
Line 367: change “pf<0.02” to “p<0.02”.
Lines 405, 407: change “1 x 109” to “1 x 109 (superscript)”.
Supporting Material Figure S1: change “10 subjects randomized to <<B+BL>> group” to “12 subjects randomized to <<B+BL>> group”.
Reviewer 2 Report
The paper submitted by Ghini and colleagues describes the alteration in human urinary and serum metabolic NMR profiles upon the intake of two different types of probiotics for the period of eight weeks. The data were subjected to multivariate and univariate statistical analyses. The authors show that probiotics supplementation has significant effect on the overall metabolome of the subjects and appears to reduce inter-individual variation in metabolic phenotypes – both interesting and important findings. The paper is generally well-written, is sufficiently descriptive, it provides adequate citations, and the quality of the figures is good. However the paper lacks a deep investigation of metabolic profiles that would reveal statistically significant metabolites.
I suggest the publication of this work, although there are some concerns regarding metabolite analysis and the interpretation of results that authors may need to comment.
- Authors only analysed “well defined and isolated” peaks of metabolites observed in the NMR spectra using univariate statistical analysis to see which of these metabolites were significantly different between three phases of intervention. Did the authors investigate the multivariate models that they generated for any additional (and possible not so easily assigned) metabolite differentiating various phases of the study? Were any of the “targeted” metabolites used for univariate analysis confirmed in multivariate analysis?
- For serum metabolomics NMR analysis, the authors used both CPMG and 1D NOESY pulse sequences. It’s unclear from the text which datasets were used for statistical analyses by multivariate approaches and if the signals of “targeted” metabolites were integrated in 1D NMR or CPMG profiles (the latter were reported to be unsuitable for quantification due to the high variability of T2 values for different small molecules).
- Did the participants of the study maintain the same diet and drug intake during the phase II and III of the study compared to the phase I (baseline)?
Reviewer 3 Report
The paper “Effects of probiotics administration on human metabolic phenothype” describes the effects of the use of probiotics on human serum and urine following an NMR-metabolomics approach. The paper is well written, and the experimental design is sound and well defined. Given the degree of complexity that could arise when it is studied human metabolic profiles, I considered the NMR-approach an adequate choice. The discussion and conclusions are assertive and the authors didn’t try to drawn any speculative interpretations beyond the observed results.Thus, I suggest acceptance for this manuscript. In any case, I pointed some corrections that I consider to be essential prior to publication.
Line 82: When it reads “urines and sera metabotypes”; I think it is more correct to write “urine and serum metabotypes”.
Line 91: When it reads “urines and sera spectra”; I think it is more corrector o write “urine and serum spectra”.
Line 158: delete the comma
Line 241 and Table 2: Here it reads “n-methyl-nicotinamide”. For consistency, select one of the possible names for the compound and keep it along the entire paper. Along the table 2, when it reads “1-methylnicotinammide”; it should reads 1-methylnicotinamide or n-methyl-nicotinamide.
Line 264: Citrulline doesn’t appear in table 2 with this designation. Again for consistency, use the same name for this compound along the entire paper.
Line 287: You refer that pyruvate increased during both phase II and phase II. However, in the table 3 there is only one arrow in pyruvate row (phase I vs phase III). Moreover, p value for phase I vs phase II is 0.078, so non significant. Please, clarify this sentence.
Line 288 – line 292 and Table 3: It is described results from L+BL and B+BL groups but these results are not showed in table 3. Please clarify or complete the table.
Line 349:Lactobacillus acidophilus should be written in italic.
Line 367: Where it reads “pf<0.02”; it should reads p<0.02. I don’t agree with the use of adjectives regarding p-values such as “most significant”, since the values are significant or not depending on your v-value chose. In this case I think you mean that even if you consider p<0.02, you can still rejected the null hypothesis that I assume is “there is no difference between the experimental groups”.
Line 405: L. delbrueckii subsp. delbrueckii should be written in italic (except subsp. word).
Line 408 – line 410: B. longum should be written in italic.
Line 424: I would like to see information about the NMR probe and the NMR tubes used for the acquisition.
Line 431: Where it reads “doubled”, it should reads doublet.
Line 455: You should add a space between “(FDR):” and “an”.
Line 458: The word “flowchart” is repeated.
Figure S1: Beginning from the top of the scheme, after the bifurcation of the arrow, we have 2 boxes. On the box of the right side, where it reads “10 subjects randomized to “B+BL” group, I think the correct number is 12.
